# A Comparative Study of the Effectiveness of Pharmacopuncture Therapy for Chronic Neck Pain: A Pragmatic, Randomized, Controlled Trial

**DOI:** 10.3390/jcm11010012

**Published:** 2021-12-21

**Authors:** Kyoung-Sun Park, Suna Kim, Changnyun Kim, Ji-Yeon Seo, Hyunwoo Cho, Sang-Don Kim, Yoon-Jae Lee, Jinho Lee, In-Hyuk Ha

**Affiliations:** 1Jaseng Hospital of Korean Medicine, Seoul 06110, Korea; lovepks0116@gmail.com (K.-S.P.); jasengjsr@gmail.com (J.L.); 2Daejeon Jaseng Hospital of Korean Medicine, Daejeon 35262, Korea; tnsdk2648@hanmail.net (S.K.); kcn1020@jaseng.org (C.K.); 3Bucheon Jaseng Hospital of Korean Medicine, Bucheon 14598, Korea; wowpan21@gmail.com (J.-Y.S.); goodsmile8119@gmail.com (Y.-J.L.); 4Haeundae Jaseng Hospital of Korean Medicine, Busan 48102, Korea; kamui0328@gmail.com (H.C.); donmuta@naver.com (S.-D.K.); 5Jaseng Spine and Joint Research Institute, Jaseng Medical Foundation, Seoul 06110, Korea

**Keywords:** neck pain, pharmacopuncture, acupuncture, physical therapy, randomized controlled trial, pragmatic clinical trial

## Abstract

Background: This two-arm, parallel, pragmatic, multicenter, clinical randomized, controlled trial with a 12-week follow-up period aimed to compare the effectiveness of pharmacopuncture therapy and physical therapy strategies for chronic neck pain. Methods: Eight sessions of pharmacopuncture therapy or physical therapy were administered within 2 weeks. The primary outcome was the visual analogue scale (VAS) score for neck pain. The secondary outcomes were the scores of the Northwick Park questionnaire (NPQ), VAS score for radiating arm pain, numeric rating scale (NRS) for neck and arm bothersomeness, neck disability index (NDI), patient global impression of change (PGIC), 12-item short form health survey (SF-12), and EuroQoL 5-dimension 5-level (EQ-5D-5L) instrument. The protocol was registered with Clinicaltrials.gov (NCT04035018) and CRIS (KCT0004243). Results: We randomly allocated 101 participants with chronic neck pain to the pharmacopuncture therapy (*n* = 50) or physical therapy group (*n* = 51). At the primary endpoint (week 5) the pharmacopuncture therapy group showed significantly superior effects regarding VAS score for neck pain and arm bothersomeness, NRS for neck pain, NDI, NPQ, and PGIC compared with the physical therapy group. These effects were sustained up to 12 weeks after follow-up. Conclusion: Compared with physical therapy, pharmacopuncture therapy had superior effects on the pain and functional recovery of patients with chronic neck pain.

## 1. Introduction

Neck pain is a common musculoskeletal disorder that affects daily routine. It causes disability in work or sports [1] and increased medical costs. According to the 2010 Global Burden of Disease, neck pain is the fourth major cause of disability after back pain, major depressive disorder, and other musculoskeletal disorders [2]. A systematic analysis of neck pain conducted in 2017 reported that the age standardized rates for point prevalence of neck pain per 100,000 population was 3551.1, for incidence of neck pain per 100,000 population was 806.6, and for years lived with disability from neck pain per 100,000 population was 352.0 in 2017. These estimates did not change significantly between 1990 and 2017 [3]. Additionally, neck pain is more common in women than in men [4,5,6]. First-line treatments for neck pain include injection therapy and oral medications. Nonsteroidal anti-inflammatory drugs (NSAIDs) have been proven effective [7]; however, they may cause adverse events, including gastritis, gastric ulcer, gastrointestinal bleeding, and myocardial infarction [8]. Moreover, the effect of non-pharmacological treatment, including dry needling, low-level laser, electrotherapy, ultrasound, and traction, is still controversial [9].

Pharmacopuncture therapy (PPT), which combines acupuncture and herbal medicine, involves the administration of herbal medicine extracts to acupoints. PPT maximizes and extends the effects of accessing acupoints based on physical and chemical stimulation [10,11]. A study conducted in 2016 investigated the frequency and details of using PPT at 12 Korean medicine (KM) hospitals and clinics. Among 33,145 inpatients and 373,755 outpatients who visited a KM hospital or clinic during a 4-year period, 32,947 (98.6%) inpatients and 289,860 (77.6%) outpatients received PPT [12]. A survey conducted in 2015 found that 95.9% of KM physicians specializing in musculoskeletal disorders used PPT for lumbar intervertebral disc displace. Especially, bee venom PPT was regarded to be most influential in the short-term treatment (8 weeks), followed by acupuncture and herbal medicine [13].

A systematic review and meta-analysis of the effectiveness of PPT for cervical spondylosis reported that PPT alone or combined with acupuncture had significantly better effects on neck pain severity and quality of life (QoL) than acupuncture alone or physical therapy (PT) [14]. However, randomized, controlled trials (RCTs) of PPT have high heterogeneity and low evidence levels; thus, definitive conclusions regarding effectiveness on neck pain cannot be derived. Therefore, a pragmatic RCT reflecting the clinical settings in Korea was performed to compare the effectiveness of PPT and PT strategies for chronic neck pain. We hypothesized that compared the standard treatment of PT, PPT for chronic neck pain would have superior effectiveness.

## 2. Materials and Methods

### 2.1. Study Design and Setting

This two-arm, parallel, multicenter RCT included 101 patients from Jaseng Hospital of Korean Medicine, Daejeon Jaseng Hospital of Korean Medicine, Bucheon Jaseng Hospital of Korean Medicine, and Haeundae Jaseng Hospital of Korean Medicine. Due to the site differences in the patient enrollment rates, the study protocol was modified (version 1.4) and a competitive enrollment method was applied. The study protocol follows CONSORT guidelines [15], and was approved by the Institution Review Board of each institution (approval no: JASENG 2019-06-009-004) before patient enrollment. The protocol was registered with Clinicaltrials.gov (NCT04035018) and CRIS (KCT0004243) and constantly renewed based on the study status. Information regarding the research institution and principal investigator is available through the trial registration site. Additionally, the protocol was previously published [16].

### 2.2. Participant Timeline

The participants completed an informed consent form (ICF) during the first visit after receiving explanations regarding the study from the investigator. Subsequently, potential participants were screened based on the inclusion and exclusion criteria for study enrollment. Starting from the second visit, participants received treatment (PPT or PT) according to their randomly allocated group. Participants attended 13 visits, including 8 treatment sessions (2 sessions per week for 4 weeks). Subsequently, follow-up observations were made through in-person visits or telephone interviews at 5, 8, and 12 weeks after the baseline examination. Appendix A shows the enrollment schedule, interventions, and participant assessments.

### 2.3. Inclusion and Exclusion Criteria

#### 2.3.1. Inclusion criteria

(1)Non-specific neck pain for more than 6 months;(2)Visual analogue scale (VAS) score > 5 for neck pain;(3)Age 17 to 70 years;(4)Provision of written informed consent.

#### 2.3.2. Exclusion Criteria

(1)Cancer migration to the spine or spinal fracture;(2)Progressive or severe neurologic deficits;(3)Cancer, fibromyalgia, rheumatoid arthritis, or gout;(4)Stroke, myocardial infarction, kidney disease, dementia, diabetic neuropathy, or epilepsy;(5)Using steroids, immunosuppressants, or psychotropic medications;(6)Hemorrhagic disease, severe diabetes, or using anticoagulants;(7)Use of NSAIDs or pharmacopuncture performed within the past week;(8)Pregnancy or lactation;(9)Cervical surgery within the past 3 months;(10)Participation in another clinical trial within 1 month or planning to participate in another trial during the follow-up period of the present trial;(11)Failure to provide written informed consent;(12)Other difficulties participating in the trial according to the investigator’s decision (who cannot read or understand the questionnaire).

### 2.4. Interventions

#### 2.4.1. Experimental Group: PPT

Participants in the PPT group were scheduled to undergo two PPT sessions per week for 4 weeks; however, one session could be added or removed per week based on the patient’s condition. Accordingly, the participants could undergo one to three treatment sessions per week. Specifically, during the earlier stages when symptoms were severe, three treatment sessions per week could be required. However, when symptom improvement occurred, only one treatment session per week could be required. The total number of treatment sessions was not predefined. Based on a previous study [12], the KM physician made treatment decisions according to the patient’s condition. To ensure accurate assessments, all intervention-related details were recorded in the medical charts. Furthermore, the PPT strategies applied were retrospectively chart-reviewed and recorded in the case report form (CRF) for analysis.

#### 2.4.2. Control Group: PT

A review of data from the Korean Health Insurance Review and Assessment Service demonstrated that various combinations of PT methods (superficial heat therapy, deep heat therapy, traction therapy, electrotherapy, etc.) were used in clinical practice depending on the patient’s symptoms [17]. Accordingly, the physician chose the PT method, treated area, and treatment duration based on the patient’s symptoms, magnetic resonance imaging findings, and degree of improvement. Participants in the PT group were scheduled to undergo two PT sessions per week for 4 weeks; however, one session per week could be added or removed depending on the patient’s condition, as previously described for the PPT group. The physician determined the total number of treatment sessions based on the patient’s condition. To ensure accurate assessments, details of the PT type, PT duration, treated area, and date of treatment were recorded in the medical charts. Moreover, applied intervention methods were retrospectively chart-reviewed and recorded in the CRF for analysis.

### 2.5. Discontinuation and Dropout Criteria

Participants could be dismissed from the study during the study period for the following reasons:(1)If a participant had a disease that was undetected during the pretrial screening that could affect the study outcome;(2)If the participant or the participant’s legal representative requested study discontinuation, or if the participant withdrew consent for study participation;(3)If pregnancy was confirmed during the study;(4)If the administered intervention for neck pain caused problems for the participant;(5)Other conditions that made study participation unfit according to the decision of the principal investigator.

### 2.6. Concomitant Treatment

Patients with severe pain during the intervention or follow-up study period were allowed to seek medication or treatment for neck pain at another medical institution. In such cases, the specific details and treatment frequency were recorded in the CRF for analysis.

### 2.7. Outcomes

#### 2.7.1. Primary Outcome

##### VAS Score for Neck Pain

The VAS is used for linear measurements of pain experienced by a patient using a 100-mm line with one end indicating “no pain” and the other end indicating “unimaginable pain.” The patient marked a specific point on the scale to indicate the neck pain intensity experienced during the past week.

#### 2.7.2. Secondary Outcomes

##### Northwick Park Questionnaire

The Northwick Park quesionnaire (NPQ) is a self-administered questionnaire comprising nine items related to daily activities affected by neck pain (neck pain intensity, sleeping, numbness, duration of symptoms, carrying, reading, and watching television, working and/or housework, social activities, and driving). Each item comprises a single question and five responses regarding increasing difficulty or pain intensity. Each item was scored using a scale of 0 to 4, with higher scores indicating worse dysfunction. The total score comprised the sum of the scores of all nine items [18]. We used the Korean version of the NPQ, which was translated by Lee et al. in 2010 and has proven reliability and validity [19].

##### VAS Score for Radiating Arm Pain

Using the VAS, the patient marked a specific point to indicate the intensity of radiating arm pain experienced during the past week.

##### Numeric Rating Scale of Neck and Arm Bothersomeness

The numeric rating scale (NRS) was used to measure bothersomeness in the neck and arm area during the past week. The patients chose a number between 0 and 10 that best described their current level of bothersomeness (0 = no pain and 10 = unimaginable bothersomeness).

##### Neck Disability Index

The neck disability index (NDI) is used to determine the level of disability during daily life by dividing the total score by the number of scored items to derive an average score. It comprises 10 items scored using a scale from 0 to 5 [20].

##### Patient Global Impression of Change

The patient global impression of change (PGIC) is used for the subjective assessment of the patient’s impression of change (improvement) and comprises seven levels (1 = very much improved, 2 = much improved, 3 = minimally improved, 4 = no change, 5 = minimally worse, 6 = much worse, and 7 = very much worse) [21].

##### 12-Item Short Form Health Survey Version 2

The 12-item short form health survey (SF-12) is used to measure the health-related quality of life (QoL). It comprises 12 items in 8 categories, including physical functioning, role-physical, bodily pain, general health, vitality, social functioning, role-emotional, and mental health. It typically requires less than 5 min to complete the questionnaire, with higher scores indicating better health-related QoL. We used the Korean version of the SF-12, which has proven reliability and validity [22].

##### EuroQoL 5-Dimension 5-Level Instrument

The EuroQoL 5-dimension 5-level (EQ-5D-5L) is the most widely used indirect measurement instrument. It indirectly determines the quality weight of the specific health state using predesignated preference scores for the functional level after a multidimensional assessment of the health state. The EQ-5D-5L comprises five dimensions (mobility, self-care, usual activities, pain, and anxiety/depression), with each item assessing the level of each dimension. Weights are assigned according to the level of each dimension; moreover, the preference scores are calculated based on these weights and constants [23].

##### Drug Consumption

The types and doses of drugs (rescue medication) used for neck pain during the study period were surveyed during each visit. Moreover, the frequency of using PT, injection therapy, and other therapies other than drugs consumed were recorded.

#### 2.7.3. Adverse Events

To assess safety, hematologic tests (white blood cells, neutrophils, lymphocytes, monocytes, eosinophils, basophils, red blood cells, hemoglobins, hematocrit level, mean corpuscular volume, mean corpuscular hemoglobin level, mean corpuscular hemoglobin concentration, platelets, and erythrocyte sedimentation rate), clinical chemistry tests (T-protein, albumin, T-bilirubin, alkaline phosphatase, alanine aminotransferase, aspartate aminotransferase, and gamma-glutamyl transpeptidase levels), and immunological tests (C-reactive protein level) were performed before and after treatment for both groups.

Adverse events (AEs) refer to undesirable and unintended signs, symptoms, or diseases appearing after an intervention, including reactions without a causal relationship with the applied intervention. During this study, AEs were determined through patient-reported symptoms and the investigator’s observations; moreover, we analyzed the frequency of AEs suspected to be associated with treatment, abnormal laboratory test findings, and serious AEs. The causality between the treatment and AEs was assessed using a 6-point scale based on the World Health Organization Uppsala Monitoring Center Causality Assessment System (1 = definitely related, 2 = probably related, 3 = possibly related, 4 = probably not related, 5 = definitely not related, and 6 = unknown). Based on the Spilker classification, AEs were classified as mild (no treatment is required and the AE does not significantly impair normal life (function)), moderate (the AE significantly impairs normal life (function), may require treatment, and can resolve after treatment), and severe (the AE requires advanced treatment and may have subsequent effects).

All reported AEs were tabulated and the AE occurrence rate was calculated. Collected blood samples were immediately discarded after analysis following the standard operating procedure of the diagnostic testing team.

#### 2.7.4. Sample Size Calculation

Our null hypothesis was that there would be no between-group difference in the post-treatment pain of patients with chronic neck pain. To test this, we used an analysis of covariance (ANCOVA) for the primary analysis; the significance level (α), type 2 error (β), and statistical power were set to 0.05 (two-sided test), 0.2, and 80%, respectively. To calculate the sample size, we referred to a meta-analysis that compared the effectiveness of PPT and acupuncture or electropuncture for cervical spondylosis [14]. Although this previous study lacked the same control group as our study, it could be referenced for sample size calculations because it reported that the effectiveness of acupuncture for neck pain was equivalent [24] or superior [24,25] to that of PT. According to the previous study, the between-group difference in the final VAS score was −1.79 (95% confidence interval (CI): −2.39 to −1.19). Furthermore, based on this previous report, the effect size was calculated as f = 0.3256. A sample calculation using G*Power version 3.1.9.4 (Heinrich Heine University, Düsseldorf, Germany) revealed that 77 participants would be required. Assuming a dropout rate of 20%, we sought to enroll 100 participants.

#### 2.7.5. Enrollment

We enrolled participants through an internet press release, subway advertisement, and posters in participating institutions.

#### 2.7.6. Randomization and Allocation Concealment

Randomization sequence was created using R studio 1.1.463 (© 2021-2018 RStudio, Inc., Boston, MA, USA) statistical software and was stratified by center with a 1:1 allocation using random block sizes of 2, 4, and 6. The randomization results were sealed in a nontransparent envelope, delivered to each institution, and secured in a double-lock locker. Randomization and allocation of registration numbers of eligible participants were performed by opening the sealed envelope. The randomization number assigned to each participant was recorded in the electronic chart.

#### 2.7.7. Blinding

Because of the nature of our intervention, only the evaluator was blinded to the group allocation and involved in the intervention. The evaluator conducted the analysis in a separate space before the intervention.

#### 2.7.8. Data Collection and Management

We used an electronic CRF obtained from the Internet-based Clinical Research and Trial Management System operated by the Korea Disease Control and Prevention Agency. The standard operating procedure was distributed as a reference for study procedures, including writing the CRF, entering data, and training evaluators and investigators at each institution. We performed a query to check the range of values. Data in the electronic CRF were cleaned and locked to block access by all investigators other than the data manager.

#### 2.7.9. Statistical Methods

We assessed the demographic characteristics and treatment expectancy of participants in each group. Continuous and categorical variables were expressed as the mean (standard deviation) or median (quartile) and frequency (%), respectively, and between-group comparisons were performed using Student’s *t*-test and the chi-square test or Fisher’s exact test, respectively.

The efficacy endpoint was the between-group difference in the extent of change in continuous outcomes (NRS, VAS, NDI, NPQ, EQ-5D-5L, and SF-12) at baseline and different time points. The ANCOVA was performed with covariant factors showing significant between-group differences at baseline as covariates and groups as fixed factors. To compare differences in each outcome throughout the study period, we calculated the area under the curve (AUC) for each visit and performedh comparisons using Student’s t-test. Moreover, we used a linear mixed model to assess the trends of change between visits. The linear mixed model was performed with the mixed model for repeated measures. We compared the percentage of patients with ≥50% reduction in neck pain (NRS and VAS scores) at different time points compared with the baseline levels. Furthermore, the Kaplan–Meier survival analysis was used to compare the time until recovery (≥50% reduction in neck pain); the curves were compared using the log-rank test. A Cox proportional hazard model was used to compare the hazard ratios (HRs). Between-group comparisons of the frequency of AEs were performed using the chi-square test or Fisher’s exact test.

We performed an intention-to-treat analysis as the primary analysis and a per-protocol analysis for participants who underwent at least six treatments during the intervention period. For missing values, we applied multiple imputation for the ANCOVA and AUC analysis. Missing time points in the survival analysis were censored. All statistical analyses were performed using SAS version 9.4 statistical package (SAS Institute, Cary, NC, USA). Statistical significance was set at *p* < 0.05.

### 2.8. Ethics Approval

Before study commencement, the principal investigator submitted the protocol, CRF, ICF, and patient enrollment announcement to the Institutional Review Board (IRB) of each participating institution and obtained the necessary approval (JASENG 2019-06-008, JASENG 2019-06-009, JASENG 2019-06-010, and JASENG 2019-06-011). Any modifications to the protocol, CRF, ICF, and patient enrollment announcement were implemented after approval from the IRB. Additionally, all changes were reflected in the trial registries. To protect trial participants, all investigators were trained to follow the Declaration of Helsinki, the Korean Good Clinical Practice Guidelines, study protocol, and standard operating procedure.

#### 2.8.1. Informed Consent

Before study commencement, the investigator obtained a voluntarily signed ICF after providing sufficient explanation regarding the trial. The participant received a copy of the ICF. When collecting personal information, such as bank account information for the transportation stipend, was unavoidable, the investigator informed the participant and obtained consent.

#### 2.8.2. Confidentiality

All personal information was strictly managed under IRB supervision, and confidentiality and protection of the personal information were assured. All collected data were anonymized. Data supplied to other institutions were provided using random codes and personal information was removed.

### 2.9. Ancillary and Post-Trial Care

Participants were provided with the emergency contact information of the principal investigator and trial manager to receive necessary care in the case of medical problems or trial-related diseases, or to ask any questions during the trial period. Participants with a direct injury related to the trial could receive appropriate medical care as determined by the investigator and could be compensated for such injury, as stipulated in the trial-related insurance policy.

## 3. Results

### 3.1. Flowchart of Participants

Between September 2019 and June 2020, 263 patients were screened. Among them, 101 patients were enrolled and randomly allocated to the PPT (*n* = 50) and PT groups (*n* = 51). One participant from each group withdrew their consent before receiving treatment; moreover, one participant from the PT group was dismissed by the investigator because of an administrative error during the random allocation process. Consequently, an intention-to-treat analysis was performed using data from 49 participants in each group. Furthermore, a per-protocol analysis was performed using data from 47 and 46 participants in the PPT and PT groups, respectively, who completed at least six treatment sessions during the 4-week intervention period (Figure 1).

### 3.2. Baseline Characteristics

Significant between-group differences in baseline characteristics were observed only for credibility and expectancy of improvement and EQ-5D-5L (Table 1).

### 3.3. Treatment

The most commonly used treatments for the PPT group were Shinbaro 2, Harpagophytum procumbens, and Shinbaro 1. During each visit, 1.41 ± 0.49 types of PPT were used. The most commonly used treatments for the PT group were interferential current therapy (ICT), deep heat therapy, superficial heat therapy, laser therapy, transcutaneous electrical nerve stimulation (TENS), and extracorporeal shock wave therapy. This was consistent with frequently used PT modalities observed in the 2014 Health Insurance Review and Assessment Service National Patient Sample dataset [17]. Moreover, during each visit, 2.00 ± 0.05 types of PT were used (Appendix A).

### 3.4. Primary and Secondary Outcomes

Compared with the PT group, the PPT group experienced significantly superior effects on the scores of the VAS for neck pain, VAS for arm pain, NRS for neck pain, NDI, NPQ, and PGIC at the 5-week follow-up. These effects were maintained at the 12-week follow-up (Table 2, Figure 2). The per-protocol analysis revealed similar results (Appendix A).

The AUC analysis of the 12-week cumulative value of each outcome revealed that, compared with the PT group, the PPT group had significantly better improvements in the scores of the VAS for neck pain, NRS for neck pain, and PGIC (Table 3), similar to the findings of the linear mixed model using the mixed model for repeated measures (Appendix A).

### 3.5. Survival Analysis

Recovery was defined as ≥50% reduction in in neck pain compared to that at baseline. Figure 3A shows the VAS scores for neck pain at 11 weeks and Figure 3B shows the NRS scores for neck pain at 11 weeks. The cumulative incidence curves for recovery events were obtained for each group. The median time to recovery measured within 11 weeks was 4 weeks after randomization in the PPT group (95% CI: 3–11); however, this was not evaluated for the PT group. The number of patients who had not yet recovered at each time point is displayed under the curve in the risk table. The absolute percent difference in patients who recovered within 11 weeks was 42.9% (95% CI: 23.2–62.5%). The HRs for the number of patients with ≥50% reduction in neck pain at week 12 were 4.31 and 3.24 for the VAS (95% CI: 2.11–8.82) and NRS (95% CI: 1.78–5.90), respectively, with the PPT group showing favorable results (Figure 3).

### 3.6. Adverse Events

Two patients in the PPT group and zero patients in the PT group had AEs determined to be associated with the intervention. Both AEs in the PPT group involved localized itching at the treatment site after the first administration of PPT; however, this symptom dissipated without additional treatment after 1 day in one patient and after 2 days in the other patient. Both AEs were mild. No serious AEs were reported (Appendix A).

## 4. Discussion

In our study, PPT significantly improved pain indices, functional scale scores, and QoL at 5 weeks, compared with PT; this improvement was sustained at the 12-week follow-up visit. The minimal clinically important difference in the NRS score for neck pain is 1.5 to 2.5 [26]. We observed a mean difference of approximately 1.66 in the NRS score for pain at the primary endpoint (week 5), which exceeded the minimal clinically important difference. This indicates that compared with PT, PPT for neck pain resulted in both statistically and clinically significant improvement.

A previous RCT reported that manual therapy along with exercise is effective in the short- to mid-term than exercise for patients with chronic neck pain and upper cervical dysfunction [27]. Cervical and thoracic manual therapy together reduced neck pain and associated neck disability more effectively than cervical manual therapy alone [28]. Occlusal appliances in conjunction with NSAIDs [29], mobilisation of the upper cervical region and craniocervical flexor training [30] were also effective for orofacial pain. In regard to pharmacopuncture, previous RCT revealed that bee venom pharmacopuncture for nonspecific chronic neck pain had a significantly better effect on the VAS scores for pain and bothersomeness than NSAIDs and combination therapy [31]. Another RCT reported that *Angelica sinesis* pharmacopuncture was significantly more effective for reducing the VAS score for pain [32].

The mean age of participants in our study was late 40s, and approximately 70% of the participants were female. The mean duration of neck pain was approximately 28 months. Furthermore, approximately 40% and 60% of the participants experienced moderate and severe pain, respectively, which indicated that most patients presented with moderate to severe chronic pain. Compared with the PT group, the PPT group experienced significantly superior effects, as evidenced by the scores of the VAS for neck pain, VAS for arm pain, NRS for neck pain, NDI, NPQ, and PGIC at the primary endpoint (week 5). Moreover, the PPT group experienced a faster recovery rate than the PT group over the course of 12 weeks during the survival analysis. The AUC analysis of the 12-week cumulative value of each outcome revealed that, compared with the PT group, the PPT group showed significantly improved VAS scores for neck pain, NRS scores for neck pain, and PGIC. Specifically, compared with the PT group, the PPT group showed significantly faster improvement in neck pain that was sustained for up to 12 weeks. There were two AEs (skin symptoms at the treatment site) in the PPT group; however, they dissipated within 1 to 2 days without any additional treatment. This is consistent with a previous report that indicated a low incidence of AEs associated with acupuncture/pharmacopuncture treatment and that most AEs were not serious [33].

In our study, the most commonly used PPT types were Shinbaro 2, Harpagophytum procumbens, and Shinbaro 1. This is consistent with the trends observed in surveys of clinicians treating cervical disc herniation [34] and lumbar spinal stenosis [35] at spine speciality hospitals in Korea. Shinbaro 2 comprises nine crude herbs (Paeonia lactiflora Pall. [Paeoniaceae], Cibotium barometz [L.] J. Smith. [Dicksoniaceae], Saposhnikovia divaricate Schiskin [Umbelliferae], Eucommia ulmoides Oliver [Eucommiaceae], Acanthopanax sessiliflorum Seem [Araliaceae], Achyranthes japonica Nakai [Amaranthaceae], Scolopendra subspinipes mutilans L. Koch [Scolopendridae], Ostericum koreanum [Maxim.] Kitagawa [Apiaceae], and Aralia continentalis Kitagawa [Araliaceae]) and is used to treat inflammation or pain in patients with musculoskeletal disorders [36]. Harpagophytum procumbens can effectively alleviate major clinical symptoms caused by osteoarthritis [37] and relieve acute low back pain [38]. Harpagophytum procumbens acts through interleukin and leukocyte migration to the affected joint area. Because of these anti-inflammatory and peripheral analgesic properties, Harpagophytum procumbens is effective for inflammatory diseases [39,40,41]. Shinbaro 1 (GCSB-5) comprises six crude herbs (Cibotium barometz, Saposhnikovia divaricate, Eucommia ulmoides, Acanthopanax sessiliflorum, Achyranthes japonica, and Glycine max), and five of these components are included in Shinbaro 2. GCSB-5 is widely used to treat neuropathic and inflammatory diseases, including osteoarthritis [42].

In the control group, most frequently used PT types were ICT, deep heat therapy, superficial heat therapy, laser therapy, and TENS. Electric therapy encompasses such modalities as ICT and TENS, and the reported effects of ICT include increased pressure pain threshold [43] and decreased pain sensitivity in myalgia patients [44] along with reduced swelling, and various applications to tissue and bone regeneration. Heat therapy is known to be effective for relieving various musculoskeletal pains by promoting resolution of inflammation through increased intraarticular temperature, increased muscle temperature [45], and vasodilation [46], influencing tissue healing through an increased oxygen uptake and faster catalyzed biochemical reactions. Laser therapy is known to be effective for deep tissues and structures by increasing cell metabolism, vascular permeability, and blood flow [47,48]. In this study, physicians chose types of PT according to the patients’ severity of pain and onset.

This study had several limitations. First, we could not blind the therapist and participants because of the obvious between-group differences in the interventions. To overcome this limitation, a resident blinded to the group allocation and not involved in the intervention performed the outcome evaluation to minimize bias. Additionally, the PPT types were not diverse with regard to published articles of the current status of frequently used PPT for spinal disease [34,35]. Moreover, the actual clinical settings were not accurately reflected because the study sites did not include clinics. These factors could have interfered with the pragmatic study design. Furthermore, although the therapeutic effect of PPT was sustained for up to 12 weeks, we could not determine the long-term therapeutic effects because follow-up only lasted 12 weeks. The significant difference between two groups in expectancy of improvement at baseline is also the limitation of our study. A previous study revealed that providing negative expectations may result in a lack of a treatment response on pain and disability, whereas verbally delivering positive or neutral expectations may be beneficial for pain-related measures [49].

However, this study is significant because it is the first pragmatic RCT of PPT for neck pain. Compared with previous RCTs of PPT, the pragmatic aspects of our study include the enrollment of a wide range of patients with chronic neck pain regardless of the radiological findings, lack of a specific treatment method provided by the physicians, provision of data regarding the intervention type frequently used for outpatient care [12,17], and autonomous selection and application of the intervention type to specific sites by the physicians according to the patients’ symptoms and radiological findings.

## 5. Conclusions

Compared with PT, PPT administered for 5 weeks significantly improved pain indices, functional scale, and QoL. Our findings confirmed that PPT is an effective and safe treatment strategy for neck pain. Furthermore, our findings reflect the level of evidence and grading of recommendations in KM standard clinical practice guidelines for neck pain and could inform clinical or policy decisions.

## 6. Protocol

The study protocol version is 1.6 (2020.06.09). Key protocol modifications and other changes after this report will be regularly updated on the trial registration site.

## Figures and Tables

**Figure 1 jcm-11-00012-f001:**
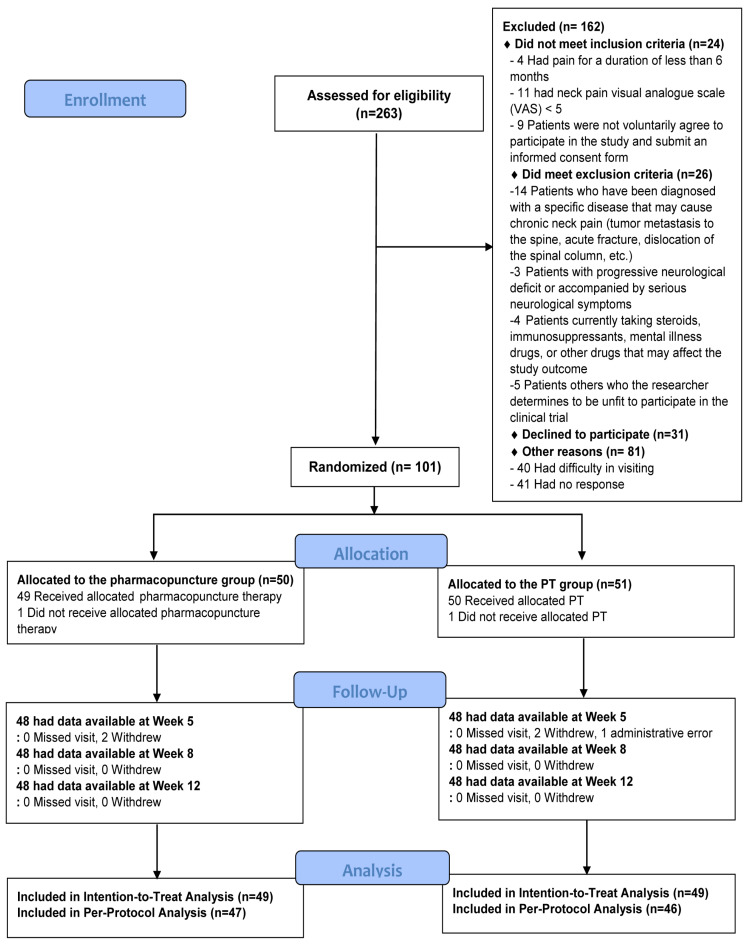
Flow chart of the study. *VAS*, visual analogue scale; *PT*, physical therapy.

**Figure 2 jcm-11-00012-f002:**
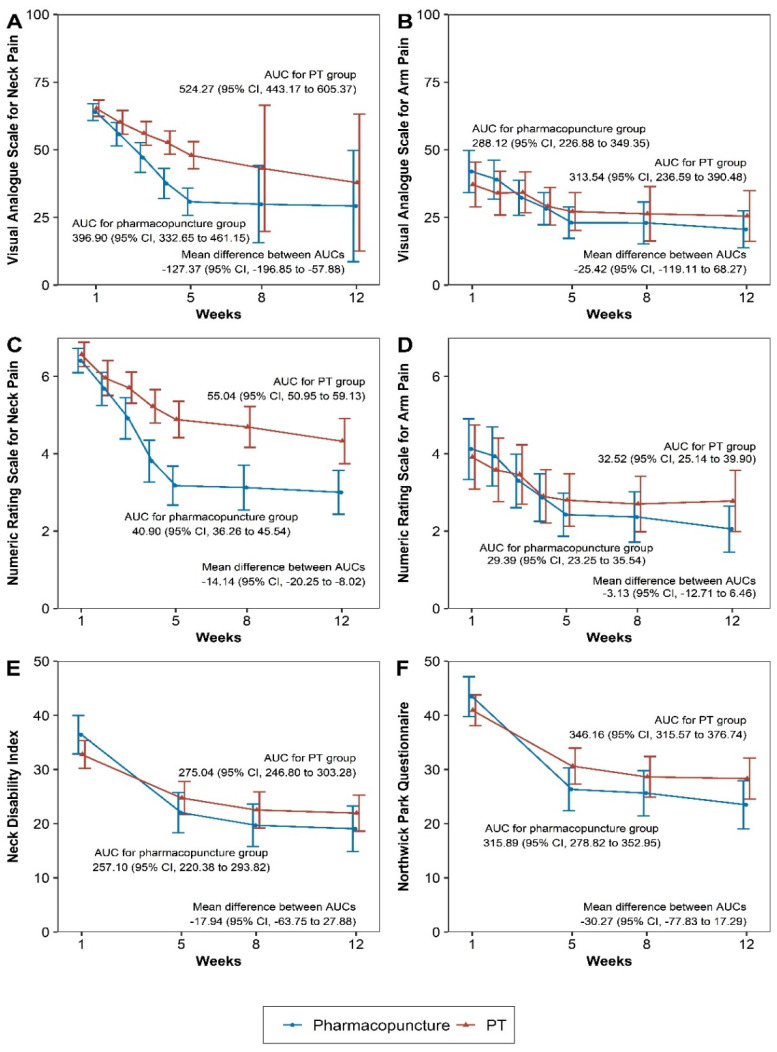
Changes in outcomes over time and areas under the curves. All graphs show the changes in scores for outcomes related to quality of life and physical function during the 12 weeks after randomization. (**A**) VAS score for neck pain; (**B**) VAS score for arm pain; (**C**) NRS score for neck pain; (**D**) NRS score for arm pain; (**E**) NDI score; (**F**) NPQ score. The dots show the mean scores and error bars show the 95% confidence intervals. Missing values were added using multiple imputation. VAS, visual analogue scale; NRS, numeric rating scale; NDI, neck disability index; NPQ, Northwick Park questionnaire.

**Figure 3 jcm-11-00012-f003:**
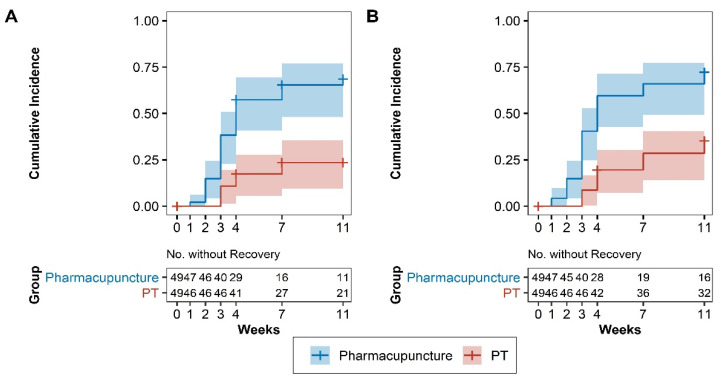
Cumulative incidence curves of recovery by group. (A) VAS scores and (B) NRS scores for neck pain at 11 weeks. A survival analysis based on the recovery of patients with ≥50% reduction in neck pain revealed that the pharmacopuncture group had a faster recovery rate than the PT group over the course of 11 weeks (*p* < 0.001, log-rank test). The HRs for the number of patients with ≥50% reduction in neck pain at week 12 were 4.31 and 3.24 for the VAS (95% CI: 2.11–8.82) and NRS (95% CI: 1.78–5.90), respectively, with the pharmacopuncture group showing favorable results. VAS, visual analogue scale; NRS, numeric rating scale; PT, physical therapy; HR, hazard ratio; CI, confidence interval.

**Table 1 jcm-11-00012-t001:** Baseline characteristics of the participants.

	Pharmacopuncture	PT	*p*-Value
	(*n* = 49)	(*n* = 49)	
Sex			
Female	34 (69.4)	35 (71.4)	0.8249
Male	15 (30.6)	14 (28.6)	
Age	49.59 ± 12.23	47.65 ± 9.88	0.3901
Height (cm)	163.20 ± 7.93	163.50 ± 8.31	0.8166
Body weight (kg)	65.08 ± 12.38	64.47 ± 11.18	0.7978
BMI (kg/m^2^)	24.39 ± 4.24	24.03 ± 3.26	0.6331
Credibility and expectancy of improvement	6.92 ± 1.32	5.61 ± 1.59	<0.0001
Duration of neck pain (months)	28.49 ± 33.30	28.37 ± 23.35	0.9832
Severity of neck pain			
Mild	1 (2.0)	0 (0.0)	0.2714
Moderate	20 (40.8)	17 (34.7)	
Severe (not requiring treatment)	11 (22.5)	19 (38.8)	
Severe (requiring treatment)	17 (34.7)	13 (26.5)	
VAS			
Neck pain	63.94 ± 11.05	65.33 ± 10.71	0.5292
Arm pain	41.96 ± 27.78	37.16 ± 29.71	0.4112
NRS			
Neck pain	6.41 ± 1.12	6.57 ± 1.12	0.4713
Arm pain	4.12 ± 2.80	3.92 ± 2.96	0.7267
NDI	36.48 ± 12.71	32.79 ± 9.09	0.1015
NPQ	43.49 ± 13.12	40.97 ± 10.13	0.2907
EQ-5D-5L	0.69 ± 0.13	0.76 ± 0.10	0.0036
SF-12			
MCS	45.48 ± 9.30	48.04 ± 9.71	0.1853
PCS	38.66 ± 8.44	40.47 ± 7.94	0.2749

Data are expressed as mean ± standard deviation (SD) or number (%). Between-group comparisons of continuous and categorical variables were performed using an independent *t*-test and the chi-square test or Fisher’s exact test, respectively. *PT*, physical therapy; Credibility and expectancy of improvement, measured using a 9-point Likert scale; VAS, visual analogue scale for pain used by patients to indicate their pain intensity using a 100-mm line (0 [no pain] to 100 [most severe pain imaginable]); NRS, numeric rating scale for pain used by patients to report their pain level as a number from 0 (no pain) to 10 (most severe pain imaginable); NDI, neck disability index calculated as a percentage, with higher scores indicating more severe disability; NPQ, Northwick Park questionnaire with scores calculated as percentages (higher scores indicated more severe pain and disability); EQ-5D-5L, EuroQoL 5-dimension 5-level instrument with scores calculated by converting the patients’ responses to a scale ranging from −0.066 (lowest quality of life) to 1 (highest quality of life); SF-12, 12-item short form health survey with scores calculated by converting the patients’ responses to a scale ranging from 0 (lowest quality of life) to 100 (highest quality of life).

**Table 2 jcm-11-00012-t002:** Primary and secondary outcomes after treatment at each time point.

		Week 5	Week 8	Week 12
VAS for neck pain	Pharmacopuncture (*n* = 49)	33.15 (27.83, 38.48)	34.07 (19.68, 48.46)	34.72 (14.17, 55.28)
	PT (*n* = 49)	17.35 (12.16, 22.55)	22.16 (−1.17, 45.49)	27.48 (2.1, 52.86)
	Difference in decrease (95% CI)	16.66 (9.9, 23.42)	12.85 (−0.57, 26.27)	8.07 (−2.4, 18.53)
	*p*-value	<0.0001	0.060	0.131
VAS for arm pain	Pharmacopuncture (*n* = 49)	18.91 (13.04, 24.79)	19.00 (10.6, 27.39)	21.38 (13.62, 29.15)
	PT (*n* = 49)	9.97 (4.1, 15.85)	10.82 (1.5, 20.13)	11.64 (2.31, 20.98)
	Difference in decrease (95% CI)	6.79 (0.22, 13.35)	5.87 (−2.07, 13.82)	7.11 (−1.5, 15.72)
	*p*-value	0.043	0.147	0.106
NRS for neck pain	Pharmacopuncture (*n* = 49)	3.23 (2.7, 3.77)	3.28 (2.68, 3.88)	3.41 (2.83, 3.99)
	PT (*n* = 49)	1.68 (1.17, 2.2)	1.88 (1.33, 2.43)	2.24 (1.63, 2.86)
	Difference in decrease (95% CI)	1.66 (1, 2.32)	1.51 (0.75, 2.27)	1.27 (0.49, 2.04)
	*p*-value	<0.0001	0.000	0.001
NRS of arm pain	Pharmacopuncture (*n* = 49)	1.70 (1.12, 2.27)	1.76 (1.1, 2.42)	2.07 (1.41, 2.72)
	PT (*n* = 49)	1.12 (0.57, 1.67)	1.22 (0.65, 1.79)	1.14 (0.33, 1.95)
	Difference in decrease (95% CI)	0.49 (−0.12, 1.1)	0.45 (−0.27, 1.17)	0.82 (−0.01, 1.65)
	*p*-value	0.117	0.217	0.053
NDI	Pharmacopuncture (*n* = 49)	14.40 (10.87, 17.93)	16.73 (12.73, 20.74)	17.38 (13.21, 21.56)
	PT (*n* = 49)	8.03 (5.28, 10.78)	10.25 (7.12, 13.39)	10.83 (7.46, 14.20)
	Difference in decrease (95% CI)	4.83 (0.82, 8.85)	4.73 (0.14, 9.31)	4.75 (−0.08, 9.58)
	*p*-value	0.018	0.043	0.054
NPQ	Pharmacopuncture (*n* = 49)	17.14 (13.42, 20.86)	17.86 (13.5, 22.22)	20.00 (15.75, 24.25)
	PT (*n* = 49)	10.32 (7.09, 13.55)	12.33 (8.75, 15.91)	12.64 (8.74, 16.54)
	Difference in decrease (95% CI)	5.70 (1.36, 10.03)	4.25 (−0.78, 9.29)	6.16 (0.99, 11.32)
	*p*-value	0.010	0.098	0.020
EQ-5D-5L	Pharmacopuncture (*n* = 49)	−0.10 (−0.13, −0.06)	−0.11 (−0.15, −0.07)	−0.13 (−0.17, −0.09)
	PT (*n* = 49)	−0.03 (−0.06, −0.01)	−0.04 (−0.07, −0.01)	−0.06 (−0.10, −0.03)
	Difference in decrease (95% CI)	−0.01 (−0.05, 0.02)	−0.01 (−0.04, 0.02)	−0.01 (−0.05, 0.03)
	*p*-value	0.371	0.640	0.500
SF-12 (MCS)	Pharmacopuncture (*n* = 49)	−4.44 (−6.98, −1.91)	−4.09 (−7.28, −0.9)	−5.80 (−9.00, −2.61)
	PT (*n* = 49)	−4.48 (−7.23, −1.73)	−4.35 (−7.19, −1.51)	−5.03 (−7.78, −2.28)
	Difference in decrease (95% CI)	1.53 (−1.53, 4.58)	1.82 (−1.82, 5.45)	0.76 (−2.81, 4.34)
	*p*-value	0.327	0.327	0.676
SF-12 (PCS)	Pharmacopuncture (*n* = 49)	−5.29 (−7.77, −2.81)	−6.68 (−9.00, −4.35)	−6.98 (−9.44, −4.52)
	PT (*n* = 49)	−1.68 (−3.84, 0.49)	−2.61 (−4.89, −0.33)	−2.82 (−5.16, −0.49)
	Difference in decrease (95% CI)	−2.63 (−5.4, 0.14)	−3.05 (−5.73, −0.37)	−3.11 (−5.88, −0.34)
	*p*-value	0.063	0.026	0.028
PGIC	Pharmacopuncture (*n* = 49)	2.39 (−2.3, 7.08)	2.52 (−2.42, 7.46)	2.35 (−2.26, 6.96)
	PT (*n* = 49)	3.10 (−2.98, 9.19)	3.13 (−3.01, 9.28)	3.15 (−3.03, 9.33)
	Difference in decrease (95% CI)	−0.71 (−0.99, −0.44)	−0.62 (−0.99, −0.24)	−0.8 (−1.12, −0.48)
	*p*-value	<0.0001	0.001	<0.0001

Effectiveness outcomes were assessed as the decrease from the baseline levels. Between-group differences were analyzed using an analysis of covariance with adjustments for the baseline values except patient global impression of change. The primary endpoint was week 5. Missing values were added through multiple imputation. Estimates for each group and between-group differences in the decrease at each time point are displayed with the 95% confidence intervals (CIs). PT, physical therapy; CI, confidence interval; VAS, visual analogue scale; NRS, numeric rating scale; NDI, neck disability index; NPQ, Northwick Park questionnaire; EQ-5D-5L, EuroQoL 5-dimension 5-level instrument; SF-12, 12-item short form health survey; MCS, mental component summary; PCS, physical component summary; PGIC, patient global impression of change. The VAS, NRS, EQ-5D-5L, SF-12, and PGIC were measured as scores. The NDI and NPQ were calculated as percentages. The PGIC was assessed using a scale from 1 (improved) to 7 (worsened), with a lower score indicating more improvement. Between-group differences were analyzed by performing an independent *t*-test of the endpoint values.

**Table 3 jcm-11-00012-t003:** Areas under the curves for the 12-week outcomes.

	Pharmacopuncture	PT	Mean Difference	*p*-Value
VAS score for neck pain	396.90 (332.65, 461.15)	524.27 (443.17, 605.37)	−127.37 (−196.85, −57.88)	0.0003
VAS score for arm pain	288.12 (226.88, 349.35)	313.54 (236.59, 390.48)	−25.42 (−119.11, 68.27)	0.5949
NRS score for neck pain	40.90 (36.26, 45.54)	55.04 (50.95, 59.13)	−14.14 (−20.25, −8.02)	<0.0001
NRS score for arm pain	29.39 (23.25, 35.54)	32.52 (25.14, 39.9)	−3.13 (−12.71, 6.46)	0.5225
NDI	257.10 (220.38, 293.82)	275.04 (246.8, 303.28)	−17.94 (−63.75, 27.88)	0.4429
NPQ	315.89 (278.82, 352.95)	346.16 (315.57, 376.74)	−30.27 (−77.83, 17.29)	0.2123
EQ-5D-5L score	8.60 (8.38, 8.82)	8.78 (8.58, 8.98)	−0.18 (−0.47, 0.12)	0.2385
SF-12 (MCS) score	541.75 (518.54, 564.96)	569.44 (546.41, 592.48)	−27.69 (−60.32, 4.94)	0.0963
SF-12 (PCS) score	481.06 (462.22, 499.91)	465.86 (445, 486.71)	15.20 (−12.74, 43.15)	0.2863
PGIC score	17.11 (15.71, 18.51)	21.93 (20.42, 23.44)	−4.82 (−6.82, −2.82)	<0.0001

The area under the curve was calculated using the trapezoidal rule. Between-group differences were analyzed using independent *t*-tests. Missing values were added through multiple imputation. The area under the curve estimates in each group and between-group differences are presented together with the 95% confidence intervals (CIs). PT, physical therapy; VAS, visual analogue scale; NRS, numeric rating scale; NDI, neck disability index; NPQ, Northwick Park questionnaire; EQ-5D-5L, EuroQoL 5-dimension 5-level instrument; SF-12, 12-item short form health survey; MCS, mental component summary; PCS, physical component summary; PGIC, patient global impression of change.

## Data Availability

We plan to share our findings with the participants, healthcare professionals, and the public through the publication of this report or trial registries. Data and materials can be requested by e-mail and will be provided after consultation with the IRB.

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
