# Peer review of "A Comparative Study of the Effectiveness of Pharmacopuncture Therapy for Chronic Neck Pain: A Pragmatic, Randomized, Controlled Trial"

_jcm, 2021, doi:10.3390/jcm11010012_

Round 1
Reviewer 1 Report
This is an interesting comparative RCT assessing effectiveness of pharmacopuncture therapy for chronic neck pain. Study is well described and executed; its' design is two-arm, parallel, pragmatic, multicenter, clinically randomized, controlled trial with a 12-week follow-up period aimed to compare the effectiveness of pharmacopuncture therapy and physical therapy strategies for chronic neck pain. As authors brought up in the abstract and introduction - neck pain is very common musculoskeletal disorder that affects the daily routine and can result in disability and increased medical costs over time- so I read this manuscript with great interest. Hence, some major criticism should be addressed:
Authors list - PhD's are redundant
This section should be structuirized according to MDPI publishing policy, please see https://www.mdpi.com/journal/jcm/instructions
Also authors should include few more keywords in compliance with MeSH - this will improve availability within search engines and will eventually lead to improved citation odds
Introduction
This section is quite short, however concise - yet some important data might have been ommitted. I also recommend putting dots after square brackets throughout entire manuscript - as compliant with standard MDPI publishing and manuscript formattting policy.
L55-L63 - this copy-paste data from abstract of articles by Lee et al. and Shin et al. are redundant - please either paraphrase these informations or move the details into discussion section.
L59 and L61 - '9' is duplicated - the only latter will do
L68 - 'a well-designed' is redundant
L70 - there is a description of 'standard treatment' missing within hypothesis section - please clarify
L72-73 - also statement 'this study aimed to provide evidence that can facilitate clinical or policy decisions' is not a valid part of hypothesis - please reformulate. Please mention also what were the study expectations.
Materials and methods
L118 - 'Other difficulties participating in the trial according to the investigator’s decision' - please be more specific as this looks suspiciously biased
L138-140 - 'Accordingly, the physician chose the PT method, treated area, and treatment duration based on the patient’s symptoms, magnetic
resonance imaging findings, and degree of improvement' - this statement is not clear, also does not fit into strict RCT scheduled schemes - thus it is a major flaw of the study. Do authors mean that there were different PT methods chosen according to single physician judgement in control group? This study is heavily biased though and should not be called a Clinical Trial in this form - please add carefully relevant information. Be very dateiled and specific.
L225 - rescue medication - were there any allowed and some were not? This should be clearly elaborated here
Discusssion
This section is quite short as entire manuscript has 34 references only. Authors should discuss their work findings with more data coming not only from Korean peninsula (a lots of mostly related self-citations are included) e.g. https://pubmed.ncbi.nlm.nih.gov/30984320 https://pubmed.ncbi.nlm.nih.gov/32927858/
https://pubmed.ncbi.nlm.nih.gov/30307636/
https://pubmed.ncbi.nlm.nih.gov/32207414/
to mention just RCTs but here were much more relevant data published within last 10 years.
Also, at the end of this section study limitations should be elaborated.
Reviewer 2 Report
Thank you for this interesting report analyzing the effectiveness of PPT in patients with neck pain. Although this manuscript is mostly well structured, I would like to convey the authors some recommendations:
Introduction:
I would recommend the authors to use updated recent systematic reviews regarding the prevalence of neck pain
In line 65, the authors should tone down this confirmation. After reading the mentioned meta-analysis, this could be applicable just for neck pain derived from spondylosis and (as you stated) is based in low to very-low quality evidence. Another recent meta-analysis reported that, even if infiltration was superior to acupuncture in pain relief, infiltration showed no effects on other outcomes. 10.1093/pm/pnab188
Methods: Did the authors follow the CONSORT guidelines? Although methods are mostly transparent, please reference.
Discussion: Even if the authors recorded expectancy of improvement, there is not enough discussion about how expectations can modulate the effects of interventions. I would recommend this reference: doi: 10.1111/papr.12749. In addition, as significant differences were found between groups, this may be one potential limitation affecting the study.
Reviewer 3 Report
see the attached file

Round 2
Reviewer 1 Report
The manuscript improved since last revison, however not significantly. Authors, instead of improving Discussion section are writing elaborates in Reply to reviewer section how very pragmatic this RCT is. To sum up, Discussion needs to be re-written and expanded as most of my former remarks (same as Reviewer 3) including inorporating provided citations were not taken into account completely.
